# Plasmonic Coupled Modes in a Metal–Dielectric Periodic Nanostructure

**DOI:** 10.3390/mi14091713

**Published:** 2023-08-31

**Authors:** Victor Coello, Mas-ud A. Abdulkareem, Cesar E. Garcia-Ortiz, Citlalli T. Sosa-Sánchez, Ricardo Téllez-Limón, Marycarmen Peña-Gomar

**Affiliations:** 1Centro de Investigación Científica y de Educación Superior de Ensenada, Unidad Monterrey, Alianza Centro 504, PIIT, Apodaca 66629, Mexico; vcoello@cicese.mx (V.C.); csosa@cicese.mx (C.T.S.-S.); 2Facultad de Ciencias Físico Matemáticas, Universidad Michoacana de San Nicolás de Hidalgo, Avenida Francisco J. Múgica s/n, Ciudad Universitaria, Morelia 58030, Mexico; kareemmasud@yahoo.com (M.-u.A.A.); mgomar@umich.mx (M.P.-G.); 3School of Engineering and Sciences, Tecnologico de Monterrey, Monterrey 64849, Mexico; 4CONACYT—Centro de Investigación Científica y de Educación Superior de Ensenada, Unidad Monterrey, Alianza Centro 504, PIIT, Apodaca 66629, Mexico; rtellez@cicese.mx

**Keywords:** plasmonics, metasurfaces, gap surface plasmons, surface lattice resonances

## Abstract

In this study we investigate the optical properties of a 2D-gap surface plasmon metasurface composed of gold nanoblocks (nanoantennas) arranged in a metal–dielectric configuration. This novel structure demonstrates the capability of generating simultaneous multi-plasmonic resonances and offers tunability within the near-infrared domain. Through finite difference time domain (FDTD) simulations, we analyze the metasurface’s reflectance spectra for various lattice periods and identify two distinct dips with near-zero reflectance, indicative of resonant modes. Notably, the broader dip at 1150 nm exhibits consistent behavior across all lattice periodicities, attributed to a Fano-type hybridization mechanism originating from the overlap between localized surface plasmons (LSPs) of metallic nanoblocks and surface plasmon polaritons (SPPs) of the underlying metal layer. Additionally, we investigate the influence of dielectric gap thickness on the gap surface plasmon resonance and observe a blue shift for smaller gaps and a spectral red shift for gaps larger than 100 nm. The dispersion analysis of resonance wavelengths reveals an anticrossing region, indicating the hybridization of localized and propagating modes at wavelengths around 1080 nm with similar periodicities. The simplicity and tunability of our metasurface design hold promise for compact optical platforms based on reflection mode operation. Potential applications include multi-channel biosensors, second-harmonic generation, and multi-wavelength surface-enhanced spectroscopy.

## 1. Introduction

Periodic arrays of plasmonic nanoparticles [1,2,3] are essential structures in nanoscience and nanotechnology and, therefore, a deep understanding of their fundamental properties and applications is necessary for the development of advanced nanophotonic circuitry [4,5,6]. Unlike the metal–dielectric interface, which only supports well-known surface plasmon polaritons (SPPs) [7,8] and localized surface plasmons (LSPs) [9,10], such arrays can couple modes between the LSP of a single constituent scatterer (typically a nanoparticle) and the diffractive modes of the lattice structure, resulting in particle–lattice field enhancements larger than those of the same number of isolated elements [11,12,13]. Such coupled electromagnetic modes are referred to as surface lattice resonances (SLRs) [2,14], and their optical properties are highly dependent on the lattice periodicity, scatterer morphology, nature of the metal, and the refractive index of the environment. Properly designed periodic nanoparticle arrays can control plasmonic dispersion curves [15,16], obtain exceptionally high Q resonances [17], and achieve subwavelength control of nano-optical fields [18,19]. Examples of such designs include nanoantennas [20], waveguides [21,22], metalenses [18,23], and nanocavities [24]. However, achieving precise control of hybridized plasmonic resonances is challenging due to the relatively high number of parameters involved in an SLR-coupled system [25,26,27]. In this context, gap surface plasmon metasurfaces (GSPMs) [28,29,30,31] based on metal–insulator–metal (MIM) nanostructures [32,33,34,35] have exhibited strong broadband absorption of multi-spectral coverage. The thickness of the dielectric function of the sandwiched material closely correlates with such absorption [36], and this geometry has also shown excellent performance in optical phase, amplitude, and polarization manipulation of reflected fields [37]. In addition to the well-established plasmonic metals, recent advancements, in this context, have introduced exciting alternatives. For example, ultra-wideband fractal metamaterials, composed of nickel and operating across the UV to IR spectrum, present unique absorption capabilities [37]. Additionally, percolation aluminum and silicon nanofilms serve as ultra-thin perfect light absorbers [38]. Furthermore, innovative plasmonic materials, such as vanadium nitride-based metasurfaces, show great promise in efficient absorber/emitter applications within solar-thermophotovoltaic systems [39]. Moreover, copper plasmonic metamaterial glazing offers directional thermal energy management functionalities, expanding the scope of plasmonic applications [40].

In this work, we report on a 2D-GSPM designed to generate simultaneous multi-plasmonic resonances, with the possibility to tune them within the near-infrared domain. Reflectance spectra of the proposed structure were calculated for different array periods, and two dips were observed with reflectivity values that nearly reached zero. The broader dip is associated with the GSP effect, and the dips with the smallest bandwidth exhibit a nearly linear dispersion in wavelength when the period of the grid is varied. The resonance associated with the GSP effect is shown to exhibit a blue shift with increasing gap thickness, while a spectral red shift is observed for gap thicknesses larger than 100 nm. We calculated the near-field optical distribution around the nanoblocks to gain a deeper understanding of the optical response of the system. This development has potential applications in multi-channel biosensors [41], second-harmonic generation [42], multi-wavelength surface-enhanced spectroscopy [43], and other interesting fields.

## 2. Materials and Methods

We used the finite difference time domain method (FDTD) to perform an electromagnetic simulation of a GSPM. The FDTD method is a numerical technique used to solve electromagnetic problems by dividing space and time into a finite number of small cells. Here, we used full 3D FDTD calculations to model the reflectance of the plasmonic system. All calculations were performed using the commercial simulation program ANSYS Lumerical, version 2016a [44].

By applying such a numerical method to the GSPM, we were able to accurately model the electromagnetic interactions within the structure and gain insights into its optical properties. This allowed us to investigate the behavior of the GSPM under different external conditions. The GSPM is composed of three layers: a bottom metal layer, a middle insulator layer, and a top layer consisting of square gold nanoblocks (nanoantennas) (Figure 1b). The sample consists of several structural parameters, including lattice periodicity (Λ), dielectric gap (*t*), nanoblock length (*L*) and height (*h*), and an infinitely thick metal substrate thickness. To simulate an infinite array, we applied a periodic boundary condition at the walls perpendicular to the unit cell, and a uniaxial anisotropic perfectly matched layer (PML) was applied in parallel to the walls. The refractive index of air at room temperature (T = 300 K) was used as the background, and the dielectric constants of gold were extrapolated from Johnson and Christy’s experimental results [45], while the optical constants of the SiO_2_ gap were extracted from Palik experimental data [46]. To absorb energy flowing at grazing incidence, we set the number of PML layers to 12. We used a mesh size of (5 × 5 × 5) nm^3^ in space and (2 × 2 × 2) nm^3^ around the nanoparticles. The simulation was run with a time step of 0.0095 fs, corresponding to a stability factor of 0.5, and a simulation time of 1500 fs. Finally, we modeled the incident light as a Gaussian wave packet composed of plane waves with the wave vector parallel to the surface and with the electric field polarized along the *x*-axis (TM polarized), as shown in Figure 1a. The incident light was normal to the surface of the structure. To calculate the reflectance, using FDTD, a frequency-domain field and power monitor is positioned behind the source injection plane. When both incident and reflected fields are present in front of the source, only the reflected fields are captured when the monitor is situated behind the source. By configuring the amplitude of the source to 1 in the software, there is no need to further normalize the power detected in the monitor, and the measured power corresponds to the reflectance of the plasmonic structure.

## 3. Results and Discussion

First, we calculated the reflectance spectra of the GSPM structure under normal incidence and for different array periods (Figure 2).

The thickness of the gold film was fixed at 100 nm, which is sufficient to achieve nearly zero optical transmission. The parameters *L* and *h* were set to 200 nm and 50 nm, respectively, while *t* was fixed at 25 nm [Figure 1b]. These parameter values align with those used in experimental studies on light–matter interactions in similar systems, as documented in the literature [47]. It is, however, worth mentioning that the values of the parameters *L* and *h* govern various aspects of the optical response, such as resonance position, strength, and the profile of a spectral peak, making them critical parameters to consider when designing plasmonic nanostructures. The length of the nanoantenna dictates the specific resonance wavelength of the LSP mode that the nanoantenna can sustain. Longer nanoantennas exhibit resonances at longer wavelengths, while shorter ones resonate at shorter wavelengths. The height of the nanoantenna cell affects the degree of coupling between neighboring nanoantennas in the lattice. Taller nanoantennas may experience stronger coupling, leading to the hybridization of resonant modes and the emergence of additional resonances in the reflectance spectrum. On the other hand. the thickness of the insulating layer in the GSPM also impacts the reflectance spectrum. Careful selection of the insulator thickness enables tuning of the resonance position, strength, line shape, and even the induction of Fano resonances. This parameter plays a critical role in optimizing and engineering the GSPM to achieve specific optical functionalities for applications in nanophotonics, sensing, and other fields.

Each calculated reflectance spectrum exhibited two distinct dips, as depicted in Figure 2. The reflectivity values at these dips approached zero, indicating the presence of resonant modes within the system. Notably, the broader dip at 1150 nm was consistently observed in all calculated spectra, regardless of the array periodicity. This dip was located far from the SPP resonances that could be generated by the grating resonance of the nanoantenna array (i.e., Rayleigh anomalies).

Hence, the origin of the reflectance suppression in our GSPM structure can be attributed to a Fano-type hybridization mechanism. This mechanism arises from the spectral overlap between the LSP resonances of the metallic nanoblock and the SPP resonances of the underlying metal layer. As a result of this coupling, a GSP resonance is generated within the system. The GSP mode is localized between the metallic nanoblock and the underlying metal layer, creating a distinct resonant state. This resonance corresponds to specific frequencies or wavelengths at which the system exhibits a significant reduction in reflection.

Consequently, there is a strong field enhancement and confinement in the dielectric region of the gap. GSP modes [28,29,30,31] exhibit broad resonances that are minimally affected by the array periodicity. This behavior arises due to the enhanced near-field coupling between the metal film and the nanoblocks when the gap thickness is much smaller than the wavelength of the incident light. This effect can be understood in the context of a Fabry–Perot resonator [13], which involves multiple reflections between metal surfaces and results in the generation of transmission peaks and reflectance dips in the system’s spectral response due to interference phenomena between the reflected waves. However, it is important to note that the GSP resonance relies on the interference of surface waves, rather than the interference of waves between two mirrors as in the case of a Fabry–Perot resonator. The detailed mechanism underlying the transition from Fabry–Perot resonance to GSPR falls outside the scope of the present study. Interested readers are referred to relevant literature, including [13] and references therein. On the other hand, the dips with the smallest bandwidth, as shown in Figure 2, exhibit a nearly linear dispersion in wavelength when the lattice period is varied. In all cases, the minimum value of the reflectance spectrum essentially coincides with the period of the grid. When light is incident on this array, one of the diffracted waves can travel along the surface at a grazing angle and interact with many other nanoantennas. This results in the creation of a mixed (hybrid) mode of LSP vibrations combined with the diffracted grazing wave, leading to an increase in the intensity of the LSP resonances. As a result of this process, the incident beam transfers energy into the LSP modes in a narrow wavelength range near a Wood anomaly [48]. In the case of the nanoantenna array, the Wood anomaly enhances the interaction of the incident light with the LSP modes, producing sharp plasmon resonances.

In addition, we examined the reflectance spectra of a GSPM structure with a constant array period but varying gap thickness, as illustrated in Figure 3.

The results showed that the resonance associated with the GSP effect experienced a blue shift with an increase in gap thickness. This blue shift can be attributed to a decrease in the effective refractive index of the GSP mode as the gap thickness increased. In other words, the change in the effective refractive index of the GSP mode, influenced by the gap thickness, caused the resonance to shift to higher frequencies resulting in the observed blue shift. However, a different trend appears with thicknesses larger than 100 nm where a spectral red shift is observed. It has been mentioned that the physical phenomena that are involved in the generation of SLR are vast. Despite this complexity, it is possible to assume, at least in a general form, that the generation of hybrid plasmonic modes arises mainly from the interaction between localized and delocalized modes, and a mutual combination of them. In this case, the resonance behaviors associated with the diffraction gratings (SPP Bloch modes) and those that have a possible origin in a GSP effect are far apart from each other, and then it is difficult to assume that they are effectively interacting (Figure 3). It is important to note that when the separation between the resonant wavelengths of two SLRs is larger than the linewidth of each resonance, there is negligible or no interaction between them. This phenomenon occurs as a consequence of the limited spectral overlap between the resonances, which is determined by the full width at half maximum (FWHM) of each resonance. It should be noted, however, that the precise criterion for determining whether two SLRs can or cannot interact is dependent on the particular system being considered and the nature of the coupling between the resonances. Another approach to describe this optical response is based on the fact that part of the scattered light can be coupled to SPPs that would exist between neighboring nanoblocks, potentially influencing the SLR of the system. The coupling of scattered light to SPPs between neighboring nanoblocks is a well-known mechanism in plasmonics. However, the extent to which this coupling occurs and its effect on the SLR of the system depends on several factors such as the geometry, size, and spacing of the nanoblocks, as well as the surrounding dielectric environment. 

The complex hybridization effects observed in our results can be better understood by analyzing the dispersion of resonance angles as the periodicity of the nanoblock arrays is varied (Figure 4). When examining the overlap of the LSPs and SLR resonance angles, an anticrossing region becomes evident. This region indicates the hybridization of modes that occur at wavelengths around 1080 nm and share similar periodicities. Within this anticrossing region, the modes exhibit a complex near-field distribution resulting from the coupling of both localized and propagating modes. This hybridization effect also explains the mechanism behind the blue shift observed when the dielectric gap thickness increases from 20 nm to 60 nm, and the red shift observed when the thickness increases from 60 nm to 180 nm (Figure 3). Changing the thickness of the gap produces a blue shift in the position of the gap LSP resonance, as expected, but as the resonance approaches the resonant wavelength of the SLR, the two modes couple and become hybridized. The coupling between these two resonances exchanges energy, as is the case of a coupled harmonic oscillator [49]. Therefore, it is not that the LSP resonance shift changes direction, but that the two resonances have exchanged their position through the coupling of both resonances when near the anticrossing zone [50]. The combination of different parameters yields a different size and distribution of the anticrossing zone (Figure 4), which offers a wide variety of possibilities for further designs with different optical characteristics. For example, by changing the periodicity of the nanoblocks (Figure 2), the SLR and LSP resonance appear far from each other, i.e., outside the anticrossing zone, and only the SLRs experience a red shift. The LSPs remain almost unperturbed, and do not experience a significant change, although the LSP resonant wavelength will eventually be modified for periods larger than 950 nm, as these two will also start to couple and hybridize (Figure 4). This coupling gives rise to unique electromagnetic phenomena, leading to enhanced light–matter interactions and intricate energy transfer mechanisms. Calculating the near-field zone around the nanoblocks can provide valuable information about the electromagnetic field distribution and can help to better understand the optical response of the system. The near-field zone refers to the region in the immediate vicinity of the nanoblocks where the electromagnetic field is strongly influenced by the presence of the nanoblocks. By calculating the near-field zone, one can determine the strength and spatial distribution of the electromagnetic field around the nanoblocks, and how it varies with changes in gap size. This information helps to validate the assumption that scattered light can be coupled to SPPs between neighboring nanoblocks, as well as provide insights into other phenomena such as plasmon hybridization and field enhancement (Figure 5). The calculated near-field optical images showed a clear field redistribution (Figure 5) once the gap becomes very large. As the gap increases, the near-field coupling tends to be weak and the GSP resonance is hardly supported; therefore, the effects of the SPPs between neighboring nanoblocks will increase, causing the spectral red shift.

## 4. Conclusions

We studied the reflectance spectra of a GSPM under normal incidence with x-polarized plane waves. The GSP effect is responsible for the broader dip centered around 1150 nm, which is common to all periods and apparently independent of array periodicity. The dips with the smallest bandwidth exhibit a nearly linear dispersion in wavelength when the period of the grid is varied. These dips are produced by the interaction of the incident light with the LSP modes, producing sharp plasmon resonances. The effective refractive index of the GSP mode changes with the thickness of the gap, resulting in a shift in the resonant frequency. By varying the dielectric gap thickness, we observed a blue shift in the resonance associated with the GSP effect for gaps smaller than 100 nm, and a spectral red shift for gaps larger than 100 nm. The transition from blue to red shift was attributed to the hybridization of localized and propagating modes in the anticrossing region, where the LSP and SLR resonances overlapped and exchanged their positions due to energy coupling. Additionally, the near-field optical distribution around the nanoblocks provided valuable information about the electromagnetic field distribution and field enhancement in the immediate vicinity of the nanoblocks. The physical phenomena that are involved in the generation of hybrid plasmonic modes arise mainly from the interaction between localized and delocalized modes, and a mutual combination of them. The exact criterion for determining whether two SLRs can or cannot interact is dependent on the particular system being considered and the nature of the coupling between the resonances. Based on these findings, potential applications for the design and optimization of plasmonic devices can be envisioned. For instance, controlling the spectral behavior of hybrid plasmonic modes through the thickness of the gap can be useful for the development of sensors, filters, and modulators. In addition, understanding the interplay between SLR and GSP modes can aid in the creation of new types of plasmonic structures with unique optical properties. Future work in this area may involve exploring the influence of other parameters on the spectral behavior of hybrid plasmonic modes, such as the material properties of the grating and the dielectric layer, the incident angle and polarization of the light, and the presence of defects or variations in the grating structure. Furthermore, investigating the potential for nonlinear optical effects in hybrid plasmonic modes may be of interest for the development of new types of all-optical switches and logic gates.

## Figures and Tables

**Figure 1 micromachines-14-01713-f001:**
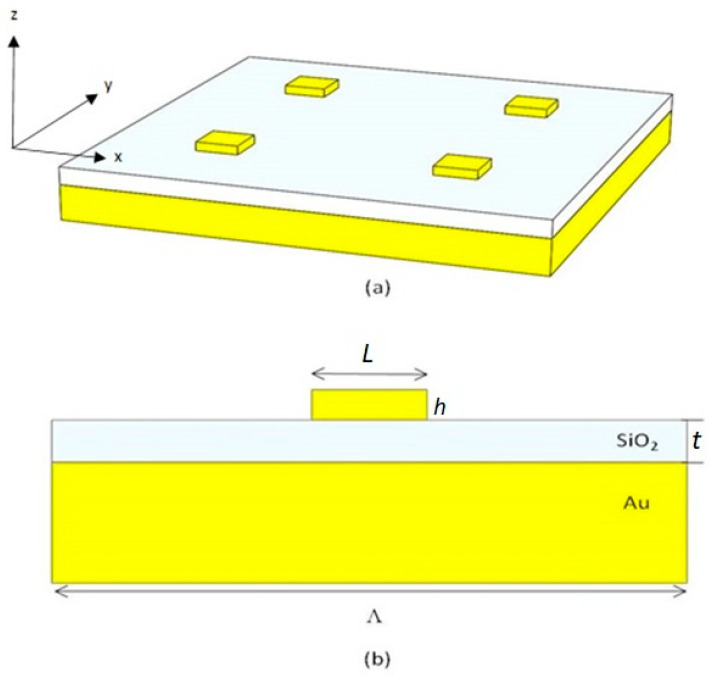
(**a**) Three- and (**b**) two-dimensional schematics of the GSP metasurface. It consists of a periodic array of nanoblocks that have been dispersed along the surface to scatter light in a controlled manner.

**Figure 2 micromachines-14-01713-f002:**
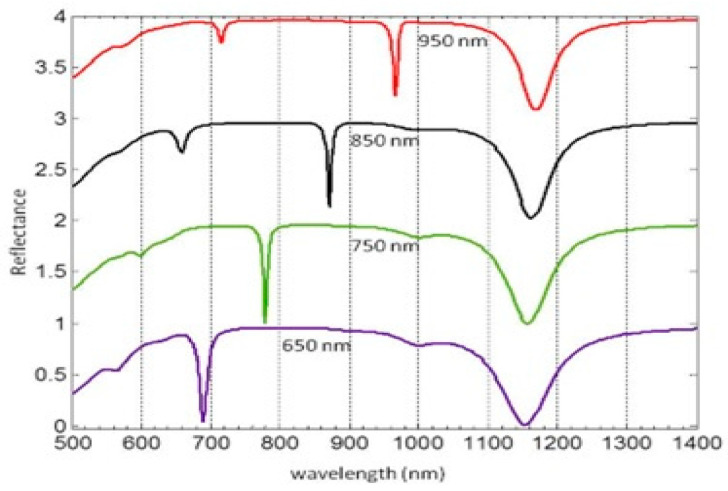
Reflectance spectra curves for different lattice periods (Λ): 950 (red line) 850 (black line), 750 (green line) and 650 (purple line) nm.

**Figure 3 micromachines-14-01713-f003:**
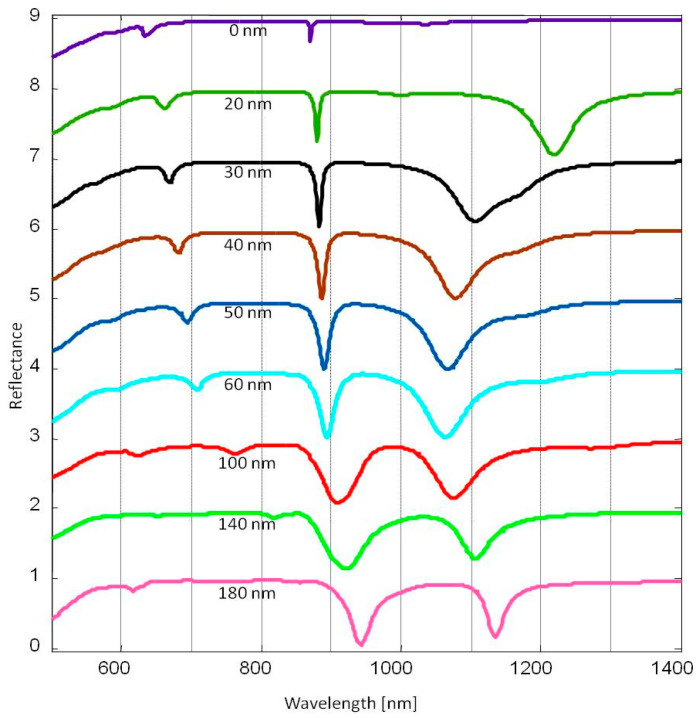
Reflectance spectra curves for different dielectric gap thicknesses. Λ was fixed to 950 nm.

**Figure 4 micromachines-14-01713-f004:**
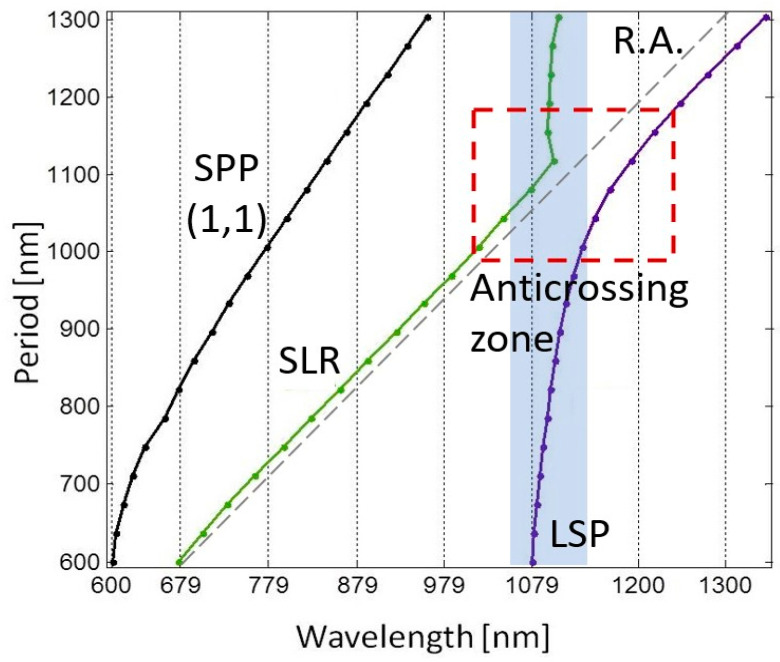
Distribution of resonance wavelengths as a function of the periodicity of the nanoblock arrays. The dispersion shows the presence of SPPs, SLRs, LSP, and an anticrossing region indicating the hybridization of localized and propagating modes. The gray dashed line corresponds to the Rayleigh anomaly (RA).

**Figure 5 micromachines-14-01713-f005:**
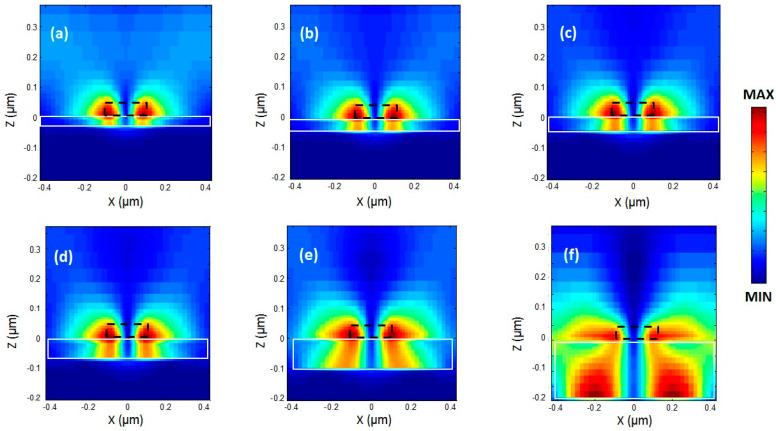
Near-field optical distributions of the resonant electric field magnitude in the (x,z) plane for different SiO_2_ layer thicknesses of (**a**) *t* = 20 nm, (**b**) *t* = 30 nm, (**c**) *t* = 40 nm, (**d**) *t* = 50 nm, (**e**) *t* = 100, and (**f**) *t* = 200 nm in a periodic GSPM array with Λ = 860 nm. The dotted (black) and solid (white) lines are to guide the eye and represent the nanoblock and the thickness of the dielectric, respectively.

## Data Availability

The data presented in this study are available on request from the corresponding author.

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
