# Peer review of "Plasmonic Coupled Modes in a Metal–Dielectric Periodic Nanostructure"

_micromachines, 2023, doi:10.3390/mi14091713_

Round 1

Reviewer 1 Report

The authors conducted a theoretical study on the reflection spectra of a GSPM under normal incidence with x-polarized plane waves. The study falls within the scope of the Micromachines Journal, and the simulation results appear to be acceptable. However, there are several issues that require attention:

1.    In the figure caption of Fig.1, " (a) 2D and (b) 1D schematic of the GSP metasurface" should be revised to "(a) 3D and (b) 2D schematic of the GSP metasurface". Please verify this correction.

2.    Please clarify whether the FDTD simulation was performed using a custom-made code or a specific software package. If a software package was used, kindly specify the name and version.

3.    Did the FDTD simulation employ a 2D or 3D model? If a 2D model was used instead of a 3D model, please explain the reasoning behind this decision.

4.    Please provide a definition for the term "reflectance" as it is used in the FDTD simulation.

5.    Line 95 mentions "reflection spectrum." Please verify if it should be "reflectance spectrum." Similarly, please review and correct instances of this term throughout the manuscript.

6.    Regarding the SPR mode with a larger wavelength in Fig.3, please provide a more detailed explanation of the mechanism behind the blueshift observed when the dielectric gap thickness increases from 20 nm to 60 nm, and the redshift observed when the thickness increases from 60 nm to 180 nm. Additionally, support your findings by citing relevant references.

7.    Please briefly describe the differences when utilizing y-polarization in the FDTD simulation.

8.    Describe the impact of the structural parameters t, L, and h on the reflectance spectrum of the proposed structure.

9.    To enhance the reader's understanding of the SLR and GSP modes arising from the proposed structure, it is suggested to include related literature, such as "J Phys D: Appl Phys, 2016, 49(47), 475102," in the references.

10. In the Introduction section, it is recommended to include several references, such as "Micromachines 2023, 14, 340," to provide readers with information on other approaches, properties, and applications of plasmonic metal-dielectric periodic nanostructures.

Reviewer 2 Report

Coello et al. proposed a surface plasmon metasurface composed of gold (Au) metal and Si substrate. Based on their numerical study they report simultaneous multi plasmonic resonances generation in IR band. Overall quality is satisfactory but before acceptance, I would suggest authors should revise it once. My comments/suggestions for the authors are listed below:

Ø  Referring to Figure 2: The dips are strongly dependent on the periodicity. However, the period has no effect on the second peak location, can the authors please explain the possible reason in the revised manuscript?

Ø  Abstract is very poorly written; I would suggest please re-write it keeping in mind the novelty of this work.

Ø  Different plasmonic metals acting as Fabry Perot resonators, where multiple reflections between metal surfaces, are previously reported for different applications. To enrich the literature review I would suggest authors should also discuss different other plasmonic platforms, few suggestions in this regard are 1) Nickel (doi.org/10.1364/OE.446423), 2) aluminum (doi.org/10.1364/OME.6.001032), and 3) Vanadium Nitride (doi.org/10.1016/j.mtcomm.2023.105416)

Ø  Referring to Figure 3: Authors explained the second resonance based on changes in the effective refractive index of the GSP mode, however, the third mode is showing a completely different behavior (both red and blue shifts). I think it's not just because of the effective index of the material, but some mode coupling also comes into play there, please look into this.  

Minor editing is required.

Round 2

Reviewer 1 Report

The paper has been thoroughly revised and is now ready for potential publication. However, there is one minor suggestion to consider: There are some typos found in the references, specifically in the Author names in Ref. 25 and 33 (where the abbreviation of the first name should only use one letter). Kindly ensure that these errors are corrected during the proofreading process before proceeding with publication.